# Molecular Mechanisms of Deregulation of Muscle Contractility Caused by the R168H Mutation in TPM3 and Its Attenuation by Therapeutic Agents

**DOI:** 10.3390/ijms24065829

**Published:** 2023-03-18

**Authors:** Olga E. Karpicheva, Stanislava V. Avrova, Andrey L. Bogdanov, Vladimir V. Sirenko, Charles S. Redwood, Yurii S. Borovikov

**Affiliations:** 1Institute of Cytology, Russian Academy of Sciences, 4 Tikhoretsky Av., St. Petersburg 194064, Russia; 2Radcliffe Department of Medicine, University of Oxford, John Radcliffe Hospital, Oxford OX3 9DU, UK

**Keywords:** tropomyosin mutation, congenital myopathy, muscle weakness, Ca^2+^-sensitivity of myofilaments, ATP hydrolysis cycle, muscle fibers, conformational changes, thin filament regulation

## Abstract

The substitution for Arg168His (R168H) in γ-tropomyosin (TPM3 gene, Tpm3.12 isoform) is associated with congenital muscle fiber type disproportion (CFTD) and muscle weakness. It is still unclear what molecular mechanisms underlie the muscle dysfunction seen in CFTD. The aim of this work was to study the effect of the R168H mutation in Tpm3.12 on the critical conformational changes that myosin, actin, troponin, and tropomyosin undergo during the ATPase cycle. We used polarized fluorescence microscopy and ghost muscle fibers containing regulated thin filaments and myosin heads (myosin subfragment-1) modified with the 1,5-IAEDANS fluorescent probe. Analysis of the data obtained revealed that a sequential interdependent conformational-functional rearrangement of tropomyosin, actin and myosin heads takes place when modeling the ATPase cycle in the presence of wild-type tropomyosin. A multistep shift of the tropomyosin strands from the outer to the inner domain of actin occurs during the transition from weak to strong binding of myosin to actin. Each tropomyosin position determines the corresponding balance between switched-on and switched-off actin monomers and between the strongly and weakly bound myosin heads. At low Ca^2+^, the R168H mutation was shown to switch some extra actin monomers on and increase the persistence length of tropomyosin, demonstrating the freezing of the R168HTpm strands close to the open position and disruption of the regulatory function of troponin. Instead of reducing the formation of strong bonds between myosin heads and F-actin, troponin activated it. However, at high Ca^2+^, troponin decreased the amount of strongly bound myosin heads instead of promoting their formation. Abnormally high sensitivity of thin filaments to Ca^2+^, inhibition of muscle fiber relaxation due to the appearance of the myosin heads strongly associated with F-actin, and distinct activation of the contractile system at submaximal concentrations of Ca^2+^ can lead to muscle inefficiency and weakness. Modulators of troponin (tirasemtiv and epigallocatechin-3-gallate) and myosin (omecamtiv mecarbil and 2,3-butanedione monoxime) have been shown to more or less attenuate the negative effects of the tropomyosin R168H mutant. Tirasemtiv and epigallocatechin-3-gallate may be used to prevent muscle dysfunction.

## 1. Introduction

In striated muscle, the interaction between myosin and actin is regulated by the tropomyosin–troponin complex depending on Ca^2+^ concentration. Tropomyosin (Tpm), together with troponin (TN), binds to actin and mediates actin-based regulation of muscle contraction [1,2]. The structure of Tpm is a coiled-coil dimer that provides specific bends to best match the surface of its extending strand to the contours of the actin filament. The electrostatic nature of the actin–tropomyosin interaction and the ability of tropomyosin to bend and change its flexibility explain the dynamic displacement of Tpm over the actin surface during the regulation process [1,3,4,5,6,7]. The positioning of tropomyosin closer to the outer or inner actin domains blocks or opens, respectively, the principal sites of stereospecific binding of myosin. Troponin imparts calcium ion sensitivity to the thin filament. It is believed that the shifting of the tropomyosin–troponin complex between different positions on actin filaments results in three functional states of the thin filament, balanced by calcium ions and a mode of myosin binding—the so-called «blocked» or calcium-free, «closed» or calcium-induced, and «open» or myosin-induced—depending on the conformation of TN and the myosin head [1,3,4,5,6,7,8,9]. Each functional state of the thin filaments correlates to a specific position of the Tpm strands relative to the outer and inner domains of actin [4,5,6,7].

The azimuthal shift of the Tpm strands over actin is possible due to the fact that these two proteins differ in their bending flexibility (therefore, due to variation in flexibility [6,8] or persistence length of these proteins [6,9,10,11]). At low Ca^2+^, the inhibitory subunit of troponin (I) directly interacts with F-actin in the thin filaments [12], thus switching it off [7,8] and producing an increase in its persistence length [9,10,11]. As a result, Tpm’s spring becomes compressed (its persistence length decreases [9,10,11]) and shifts to the outer domain of actin (to the blocked position) [7]. In this «OFF» state of the thin filament [7], the strong binding of myosin with actin is almost impossible [1,5,6,7]. Ca^2+^ binding to the Ca^2+^-sensitive subunit of troponin (C) induces some actin monomers to change their conformation to the «switched-on» state and the persistence length of the actin filament to decrease [9,10,11]. In turn, Tpm increases its persistence length [9,10,11] and moves towards the inner domain of actin to the closed position [4,5,6,7,12], partly exposing the myosin-binding site [7,12]. Conformational changes of the myosin heads provide the formation of a large area of binding to actin and the switching of the actin monomers on, followed by the decrease in persistence length of the actin filament and the increase of that of Tpm [8,9,10,11]. In this «ON» state, the full exposure of the myosin binding sites on actin by the Tpm strand occurs, and muscle contraction initiates [7,12,13]. Recently, amino acid residues involved in the actin–Tpm interaction have been identified [14,15]. In addition, there is an assumption that tropomyosin can bind to the myosin head, regulating the binding of the latter to actin at some stages of the ATPase cycle [14,15,16]. Therefore, tropomyosin appears to be the central link in the regulation of muscle contraction.

In skeletal muscle, there are three main Tpm isoforms, α-, β-, and γ (or α-slow)-Tpms, which are encoded by the TPM1, TPM2, and TPM3 genes, respectively [17]. All three isoforms exist either as homodimers or as heterodimers [18,19]. α-Isoform (Tpm1.1) and γ-isoforms (Tpm3.12) of tropomyosin are exclusively expressed in fast or slow fibers, respectively. β-Isoform of tropomyosin (Tpm2.2) is expressed in both types of fibers [20,21,22]. Tropomyosin forms alpha-helical coiled-coil homodimers or heterodimers consisting of one α- or γ-chain and one β-chain. These dimers polymerize head-to-tail into a continuous strand that binds along the entire length of the actin filament.

The primary structure of tropomyosin is represented by a repeating motif of seven residues *a*–*b*–*c*–*d*–*e*–*f*–*g* [23]. The residues at positions *a* and *d* in the repeat are canonically hydrophobic, creating a hydrophobic pocket between the two tropomyosin chains and facilitating the dimerization of salt bridges. The residues at positions *b*, *c*, and *f* are localized on the surface of tropomyosin dimers and likely modulate the interactions with proteins such as actin, troponin, and others.

Amino acid substitutions in tropomyosin molecules caused by gene mutations can change the regulatory properties of the tropomyosin strands, leading to dysregulation of contractile activity. Mutations in the Tpm genes give rise to a wide range of clinically, histologically, and genetically heterogeneous neuromuscular and cardiac diseases. Numerous point mutations in the TPM2 and TPM3 genes have been found in patients with congenital pathologies such as nemaline myopathy, cap-myopathy, distal arthrogryposis, and congenital muscle fiber type disproportion (CFTD), which are characterized by muscle weakness and hypotension (for reviews, see [24,25,26,27]). Recently, a number of Tpm mutations associated with congenital myopathies (K7X, K49X, K128E, R91G, etc. in TPM2; R91P/C, R168C/G/H, K169E, R245G/I, etc. in TPM3) have been shown to affect conserved residues with charged radicals along the entire length of the tropomyosin surface, which either electrostatically interact with the Asp25 residue of actin or are adjacent to such a residue in Tpm’s sequence [27]. Molecular dynamics simulations predicted that a decrease in the charge due to these substitutions and deletions could increase the distance between Tpm and the filament axis and change the position of Tpm if contacts with actin are disturbed [28]. It can lead to muscle disease. In particular, R168H in TPM3 has been detected to be associated with hypotension, feeding difficulties, motor delay, and scoliosis, requiring non-invasive ventilation while ambulant [29]. Muscle biopsies showed fiber-type disproportion with no other morphological changes in skeletal muscle tissue. Although the structure and function of tropomyosin are well established, the mechanisms by which the R168H mutation in TPM3 causes muscle weakness remain poorly understood.

Here, we studied the effect of Arg for His substitution at position 168 in recombinant Tpm3.12 on the actin–myosin interaction at various simulated stages of the ATPase cycle (±Ca^2+^) by using polarized fluorimetry, which is an adequate approach for such studies [8,9,10,11,30,31,32]. The results obtained show that the R168H substitution affects the ratio of switched-on and switched-off actin monomers, the balance of myosin heads strongly and weakly bound to F-actin, and the positions tropomyosin takes during the ATPase cycle. It was suggested that the R168H mutation changes the concerted interdependent conformational rearrangements of all major participants in the contractile system, which keeps the Tpm strands close to the open position. As a result, actin monomers are switched on at low levels of Ca^2+^, which activates strong binding of myosin heads to F-actin in the presence of MgATP, making relaxation difficult. At high Ca^2+^, the mutation inhibits the ability of myosin heads to form strong bonds with F-actin. Therapeutic agents tirasemtiv, epigallocatechin-3-gallate, omecamtiv mecarbil, and 2,3-butanedione monoxime can attenuate the damaging effects of the R168H mutation.

## 2. Results and Discussion

### 2.1. Ghost Muscle Fibers with Incorporated Labeled Protein as a Model for Studying the Conformational Changes in Proteins during Muscle Contraction

In this work, we reconstructed thin filaments in single ghost muscle fibers using exogenous Tpm, and TN decorated them with S1 (Figure 1). Despite the co-sedimentation finding about the decreased affinity of the R168H-mutant Tpm for actin alone [33], SDS–PAGE showed a high content of the mutant Tpm in ghost fibers. In a highly organized system of muscle fibers, tropomyosin with a defect is much easier to bind to F-actin in the form of an extended cable due to the cooperativity of the binding than in protein solution [34]. It is worth adding the equal affinity of the mutant Tpm to actin in the presence of troponin and Ca^2+^ [33]. After protein incorporation, we mimicked several steps of ATP hydrolysis [8,9,10]. In order to study the effect of the R168H mutation in Tpm3.12 on the behavior of the Tpm–TN system and on the response of myosin heads and actin to the Tpm movement during the ATPase cycle, we measured the polarized fluorescence of S1 labeled with a fluorescent probe—1,5-IAEDANS [8,35]. The polarized fluorescence of the studied protein reflects the average structural state of the population of its molecules [8,9,10,11]. The AM state of the actomyosin complex was simulated in the absence of nucleotides; MgADP and MgATP were used to mimic the AM^•ADP and AM*•ATP states, respectively, where M, M^, M* denote that the myosin heads are in different conformational states [8,9,10,11]. The 1,5-IAEDANS fluorescent probe covalently linked to Cys707 of S1 [35] revealed changes in the spatial arrangement and mobility of myosin heads in muscle fiber during the ATPase cycle [8,9,10,11,30,31,32].

It is clear that the modification of Cys707 with a fluorescent probe may affect some aspects of myosin behavior; nevertheless, the study of fluorescence from such labels remains an effective tool for gaining information about the actin–myosin interaction [35]. Cys707 labeling can reduce the ATPase activity of myosin and the sliding velocity of actin over myosin [36]. Furthermore, the modification has been shown to diminish the rotation of the myosin head converter region, which takes place during the ATPase cycle [36]. However, myosin heads modified with fluorescent probes retain nucleotide sensitivity. In particular, Cys707 modification by 1,5-IAEDANS does not affect strong binding (in the absence of a nucleotide or in the presence of MgADP) and weak binding (in the presence of MgATP) of S1 to actin [37]. In our control experiments, we also did not find any essential effect of the modification on either the strong or weak binding of S1 to actin [8,9,10,11]. Thus, within the framework of the experimental design, 1,5-IAEDANS-labeled S1 (S1-AEDANS) can be employed to study the actin–myosin interaction during the ATPase cycle. We used modified myosin heads in order to determine whether the mutant Tpm could affect the strong and weak binding of myosin heads to F-actin. The change in binding was assessed based on the alterations in the orientation and mobility of the myosin head [8,9,10,11,30,31,32,35,37].

### 2.2. Troponin (±Ca^2+^) Can Change Concerted Interdependent Conformational Rearrangements of Major Proteins in the Contractile System during the ATPase Cycle

Consistent with our earlier findings [8,9,10,11], the binding of S1-AEDANS to F-actin in ghost fibers initiated polarized fluorescence. When the helix plus isotropic model (see Section 4) was fitted to the fluorescence polarization data for S1-AEDANS, the angle between the fiber axis and the emission dipoles of the probe (Φ_E_), bending stiffness of F-actin (ɛ), and the relative number of disordered probes (N) of S1-AEDANS were found to be dependent on Ca^2+^ and nucleotides (Figure 2). A similar dependence of the fluorescence parameters for S1-AEDANS was obtained earlier in experiments with α-, β-, and γ-Tpms [8,9,10,11,30,31,32].

According to Figure 2, for the ghost fibers containing wild-type Tpm and S1-AEDANS (actin–WTTpm–S1-AEDANS), the values for the angle Φ_E_ between the fiber axis and the emission dipole of the probe, the bending stiffness ε, and the relative number of disordered fluorophores N were found to be equal to 44.0°, 8.0 × 10^−26^ N·m^2^, and 0.168 rel. units, respectively. This indicated that the probes were highly oriented relative to myosin heads and strongly bound to F-actin [8,9,10,11,35]. Since the fluorescent probe was rigidly bound to S1, it was assumed that the ε parameter contains information about the bending stiffness of the actin filaments in the area of localization of myosin heads, while the N parameter allows one to assess the flexibility of attachment of myosin heads to F-actin [8,9,10,11]. The bending stiffness of actin filaments (Figure 2b) did not differ much from that determined using polarized fluorescence from another fluorescent probe (for example, for FITC-phalloidin) [10,11,38,39]. Thus, the bending stiffness of actin filaments can be determined based on data obtained using a probe bound to the myosin head.

Binding of TN to the actin–WTTpm–S1-AEDANS complex induced conformational rearrangements of all major proteins in the contractile system. According to Figure 2, Ca^2+^ can modify these rearrangements, demonstrating their interdependence by changing the fluorescence parameters: at high Ca^2+^, the values of Φ_E_, N, and ε slightly decreased by 0.4°, 0.038 rel. units, and 0.9 × 10^−26^ N·m^2^ (*p* < 0.05), respectively. The observed changes in these parameters can be interpreted as showing an increase in the amount of myosin heads strongly bound to F-actin in ghost muscle fibers (decrease in Φ_E_ and N), which is accompanied by a decrease in the persistence length of F-actin in the thin filament, showing an increase in the amount of switched-on actin monomers (decrease in ε) [8,9,10,11,37]. It is known that a decrease in the persistence length of F-actin caused by the formation of strong binding of myosin to F-actin at high Ca^2+^ correlates with an increase in the persistence length of the Tpm strands and a shift of Tpm to an open position in thin filaments [8,9,10,11,38,39,40]. Therefore, it can be assumed that the information obtained using the myosin-bound probe makes it possible to judge the movement of tropomyosin relative to the inner actin domain. Thus, WTTpm at high Ca^2+^ shifts towards the open position and is able to facilitate the strong binding of myosin heads to thin filaments, with an increase in the amount of the switched-on actin monomers.

On the contrary, at low Ca^2+^, the values of Φ_E_ and ε increased by 1.3° and 0.3 × 10^−26^ N·m^2^, and the value of N did not change (*p* < 0.05) (Figure 2). Such changes in these parameters can be interpreted as a decrease in the amount of myosin heads strongly bound to F-actin in ghost muscle fibers, which is accompanied by a decrease in the amount of the switched-on actin monomers and an increase in the persistence length of F-actin in thin filaments [10,11,38,39,40]. Since an increase in the persistence length of actin filaments correlates with a decrease in the persistence length of the Tpm strands, it can be assumed that, at low Ca^2+^, Tpm shifts towards the blocked position [10,11,38,39,40]. Therefore, WTTpm at low Ca^2+^ moves towards the blocked position, inhibits strong binding of myosin heads to thin filaments, and reduces the amount of switched-on actin monomers.

It is known that myosin cross-bridges are cyclically detached and re-attached with the actin filament and, during force generation, pass through several conformational states, called «strong» and «weak» forms of myosin binding to actin [1]. According to Figure 2, the intermediate state of myosin heads induces a definite interdependent structural-functional state of actin, Tpm, and TN and a specific position of the Tpm strands in the thin filament. During the transition from the AM to the AM*•ATP state, a multistep change in the values of Φ_E_, ɛ, and N for S1-AEDANS was observed (Figure 2). In particular, the N values increased from 0.130 at high Ca^2+^ and 0.155 at low Ca^2+^ to 0.407 rel. units (Figure 2c). This increase in the N value may be the result of a decrease in the affinity of S1 for F-actin [8,35]. The ɛ values increased from 7.1 × 10^−26^ N·m^2^ to 8.0 × 10^−26^ N·m^2^ at high Ca^2+^ and from 8.3 × 10^−26^ N·m^2^ to 10.1 × 10^−26^ N·m^2^ at low Ca^2+^. An increase in the persistence length of F-actin filaments can be associated primarily with changes in the conformation of actin monomers that occur during the binding of myosin heads and the appearance of an electrostatic interaction between F-actin, myosin, and Tpm [3,6,7,10,11]. Since an increase in the persistence length of actin filaments correlates with a decrease in the persistence length of Tpm and a shift of the Tpm strands towards the blocked position in the thin filaments [10,11,38,39,40], it can be assumed that during the transition from AM to AM*•ATP state at low Ca^2+^, the Tpm strands move to the blocked position and the majority of actin monomers are most likely in the switched-off state, and myosin heads are weakly bound to F-actin. In the presence of MgADP, the Φ_E_ and ε values for S1-AEDANS were lower than the corresponding values obtained in the absence of nucleotides (Figure 2a,b). This means that the position of the Tpm strands is closer to the inner actin domains, and the number of the switched-on actin subunits and strongly bound myosin heads in the actin–S1–WTTpm–MgADP complex is less than in the complex without nucleotides.

The data obtained earlier for α-, β-, and γ-Tpms [8,9,10,11,37,38,39,40,41,42,43], as well as the results presented here (Figure 2), showed that, when modeling the ATP hydrolysis cycle, during the transition from weak to strong binding of myosin heads with actin, tropomyosin filaments lengthen, thereby opening up places for strong binding of myosin heads to F-actin. At the same time, the persistence length of actin shortens (Figure 2b), reflecting the switching-on of actin monomers [10,38,39,43]. In the ATPase cycle, each intermediate state of actomyosin corresponds to the multistep changes in the persistence lengths of actin and tropomyosin associated with a change in the position of the Tpm strands relative to the outer and inner domains of the actin monomer and the relative amount of switched-on actin monomers in thin filaments (Figure 3a–c). Since the regulation of muscle contraction is carried out by the concerted interdependent conformation rearrangements of all major proteins involved in this process, a change in the structural state of tropomyosin caused by the R168H mutation in TPM3 can cause an alteration in the structural and functional state of all major proteins of the contractile system. Indeed, such changes in myosin, actin, and tropomyosin have been found (see below).

### 2.3. The R168H Mutation Allows Strong Binding of Myosin Heads to F-Actin at Low Ca^2+^ and Reduces It at High Ca^2+^

According to Figure 2, for the actin–R168HTpm–S1-AEDANS complex, the angle Φ_E_, bending stiffness ε, and relative number of disordered fluorophores N were found to be 43.8°, 6.5 × 10^−26^ N·m^2^, and 0.180 rel. units, respectively. This indicates that in the presence of the mutant Tpm, the number of myosin heads bound strongly to F-actin slightly increased (or did not change), and the persistence length of actin filaments in the thin filaments was reduced compared to fibers containing WTTpm. Since a decrease in the persistence length of actin filaments reflecting the switching of actin monomers correlates with an increase in the persistence length of the Tpm strands, it is possible to suggest that the mutation shifts the tropomyosin strands into the open position [10,11]. Such changes in the structural and functional state of thin filaments, as a rule, lead to an increase in the number of myosin heads strongly bound with actin. Similar activation of the thin filaments by mutations in TPM2 and TPM3 was observed previously [9,11]. In contrast to the actin–WTTpm–TN–S1-AEDANS complex, in the complex in which WTTpm was replaced by R168HTpm, the number of myosin heads strongly bound to F-actin decreased at high Ca^2+^ (in the absence of nucleotides, Φ_E_ and N were higher by 0.9° and 0.086 rel. units and ε was lower by 0.8 × 10^−26^ N·m^2^ for R168HTpm than for WTTpm, respectively, *p* < 0.05, Figure 2). It is known that troponin at high Ca^2+^ activates the formation of myosin heads strongly bound to actin in the actin–WTTpm–TN–S1–AEDANS complex; in addition, the shifting of tropomyosin to the open position should also cause an additional increase in the amount of strongly bound myosin heads. However, as shown in Figure 2, at high Ca^2+^, the number of such heads decreased in the presence of R168HTpm. Actin–myosin ATPase activity assay displayed reduced potentiation of the filaments containing Tpm with the R168H substitution; that is, the switch of the ATPase from the inhibited into the activated state required much higher concentrations of S1 [33]. In single membrane-permeabilized fibers from patients, the R168H mutation decreased cooperative thin filament activation in combination with reductions in the myosin cross-bridge number [25]. Since the mutant tropomyosin retains the ability to activate the formation of strong binding of myosin to actin in the absence of troponin (see above), and this ability is inhibited in the presence of troponin and high Ca^2+^ (Figure 2 and Figure 3), it is possible that the reason for the decrease in the number of myosin heads in the strong-binding state at high Ca^2+^ is the abnormal work of troponin—inhibition of troponin’s ability to cooperatively activate strong actin–myosin binding.

A decrease in the number of myosin heads strongly bound to F-actin at high levels of Ca^2+^ was also observed in the presence of MgADP. At the mimicking AM^•ADP state, the Φ_E_ and N values were higher by 0.2° and 0.055 rel. units, and the ε value was lower by 0.6 × 10^−26^ N·m^2^ than for WTTpm (Figure 2). Upon mimicking the AM*•ATP state, the parameters Φ_E_, ε, and N changed: as compared to WTTpm, the value of Φ_E_ decreased by 3.0°, the value of ε did not change, and the value of N slightly decreased by 0.033 rel. units, *p* < 0.05 (Figure 2). Thus, the amplitude of a change in the values for Φ_E_ during the transition of myosin heads from weak to strong binding with F-actin (between weak binding in the presence of MgATP and strong binding in the absence of the nucleotides at high Ca^2+^) was 5.8° for R168HTpm (Figure 2a), which was much smaller than the amplitude observed for WTTpm (9.7°). It can be assumed that the R168H mutation inhibits the efficiency of the cross-bridge work [9,11,37,38,39,40,43]. This conclusion is consistent with data showing a decrease in the force production for single membrane-permeabilized fibers containing the R168HTpm [25] and in the fraction of moving filaments and their velocities [27,33].

On the contrary, at low Ca^2+^, R168HTpm initiated an increase in the number of myosin heads strongly bound to F-actin (in the absence of the nucleotide, the values of Φ_E_ and ε were lower by 1.7° and 1.8 × 10^−26^ N·m^2^, respectively, and the value of N was higher by 0.018 rel. units than in the absence of the mutation, *p* < 0.05, Figure 2).

A similar increase in the number of myosin heads strongly bound to actin at low Ca^2+^ was also observed in the presence of MgADP. In this case, the Φ_E_ parameter decreased by 1.5°, ε increased by 0.5 × 10^−26^ N∙m^2^, and N did not change. It is known that at low Ca^2+^, troponin inhibits the formation of strong myosin binding to actin; in addition, shifting of WTTpm to the blocked position causes an additional decrease in the number of strongly bound myosin heads. However, as shown in Figure 3, at low Ca^2+^, the amount of strongly bound myosin heads increased for the actin–R168HTpm–TN–S1-AEDANS complex. A similar increase in Ca^2+^ sensitivity in thin filaments containing the E173A-mutant Tpm was observed earlier [11]. Since the mutant tropomyosin retains the ability to activate the formation of strong myosin binding to actin in the absence of troponin (see above), and this ability is activated in the presence of troponin and low Ca^2+^ (Figure 2 and Figure 3), it can be assumed that the reason for the increase in the number of myosin heads in the strongly bound state at low Ca^2+^ is the abnormal work of troponin, that leads to activation of the formation of such conformational states of myosin heads.

In the presence of MgATP at low Ca^2+^, the R168H mutation decreased the value of Φ_E_ by 3.3° and increased the value of ε by 3.0 × 10^−26^ N·m^2^. This means that there is an increase in the amount of myosin heads strongly bound to F-actin and switching actin monomers on in F-actin. It is possible that the R168H mutation initiates the appearance of abnormal cross-bridges, making it difficult for the muscle to relax. Thus, the mutation in the presence of MgATP at low Ca^2+^ is able to activate strong binding of myosin heads to thin filaments, increase the amount of the switched-on actin monomers, and move the Tpm strands toward the blocked position during relaxation (Figure 3). Therefore, at low Ca^2+^, one can speak of increased Ca^2+^-sensitivity of the contractile system, which is consistent with the ATPase activity measurements in this and previous studies (Figure 4, [33]). However, at submaximal concentrations of Ca^2+^, we observed pure activation and a reduction in the proportion of strongly-bound myosin cross-bridges, as in previous work performed in muscle fibers [25], which is consistent with data on the defective activation of ATPase [33]. That is, it turns out that activation requires a higher concentration of Ca^2+^, and in this vein, Ca^2+^ sensitivity is a relative parameter and shows a downward trend, which explains the contradiction with the data obtained in other model systems [25,27].

It seems that the replacement of the positively charged arginine 168 by the positively charged histidine residue, apparently, cannot cause the destruction of the salt bridge [12,25,44], but can partially change the interaction of Tpm with TN [45] and actin [25]. This can lead to a decrease in the number of strongly bound myosin heads at high levels of Ca^2+^, an increase in the number of such myosin heads at low levels of Ca^2+^, and the appearance of myosin heads strongly bound to F-actin while mimicking the relaxation of muscle fibers (Figure 3). Similarly, more strongly bound myosin heads (than for WTTpm) were found by us earlier when simulating relaxation in muscle fibers containing point mutants: A155T [40] and E173A [11] in Tpm3.12, ΔE139 [9] and R91G [43] in Tpm2.2, R168H in Tpm1.1 [31]. These mutations have been associated with cap myopathy and arthrogryposis and were accompanied by contractures and disorganization of muscle fibers [26,27].

### 2.4. The Myosin Inhibitor, BDM, Attenuates the Effect of the Mutation on Troponin Function at Low Ca^2+^, but Is Unable to Repair the Damage Caused by the Arg168His Mutation

The information about critical conformational changes underlying the onset of muscle fiber dysfunction caused by tropomyosin mutations can be used in the selection of pharmacological agents to restore the contractile function of muscle tissue [11,39]. We tried to restore the contractility of muscle fibers destroyed by the R168H mutation using the data on the ways thin filaments switch on or off and the myosin heads are activated in the presence of the R168H mutation (Figure 2 and Figure 3).

2,3-Butanedione monoxime (BDM) is a noncompetitive inhibitor of the chemical and motor activities of myosin-II [46]. BDM has been shown to reverse myocardial ischemia when performing cardiopulmonary resuscitation in simulated cardiac arrest [47]. It is assumed that BDM stabilizes the converter domain of cardiac myosin in an intermediate conformational state characteristic of weak binding to actin and inhibits the rate of phosphate release from the active site, i.e., affects the kinetics of muscle contraction. The study of the binding kinetics [48] showed that the interaction between BDM and the γ-phosphate group of ATP in the ATP-binding site of myosin potentiates the cleavage stage, shifting the equilibrium from the M*•ATP state to the M**•ADP•Pi state, and then in a steric manner slows down Pi release and the formation of the M^•ADP complex [46,48]. From this, it follows that actomyosin is retained in the AM**•ADP•Pi state, that is, in an intermediate state between the weak and strong forms of binding.

Adding 20 mM BDM to a muscle fiber containing R168HTpm in thin filaments and S1-AEDANS changes the polarized fluorescence parameters of the probes (Figure 5). It has been shown that in the presence of 20 mM BDM, the number of activated actin monomers and myosin heads strongly bound to actin decreases at low and high Ca^2+^, and troponin does not switch the thin filaments on at low Ca^2+^ concentrations.

Myosin heads almost completely lose their ability to undergo conformational rearrangements because in the absence of a nucleotide and the presence of ADP, the value of Φ_E_ increases, indicating the formation of weak forms of myosin–actin binding. However, the N value, which indicates the rigidity of binding of myosin heads to actin, on the contrary, decreases. Apparently, there is an influence of BDM on the intradomain structure of myosin itself, as a result of which the mobility of the binding region becomes less mobile, although it tilts away from the axis of the muscle fiber. The main thing that we observed is that BDM turns myosin conformational rearrangements into ineffective ones since the conformation of the head changes slightly when modeling relaxation (Φ_E_ increases to 46.2° with ATP at high Ca^2+^ and up to 48.5° with ATP at low Ca^2+^ instead of the corresponding values of 53.3° and 52.3° in the presence of wild-type tropomyosin). That is, BDM does not induce an increase in the weak-binding forms of myosin to actin, which is required to correct the effect of the R168H mutation. The amplitude of changes in Φ_E_ decreases for muscle fibers containing the R168HTpm in the presence of BDM, indicating that the efficiency of the average statistical cross-bridge is reducing. BDM, therefore, does not restore the ability of troponin to switch the thin filaments on at high Ca^2+^. Thus, BDM attenuates the effect of the mutation on troponin function at low Ca^2+^, but is unable to completely repair the damage caused by the Arg168His mutation.

### 2.5. The Myosin Activator, Omecamtiv Mecarbil, Weakens the Effect of the R168H Mutation on Troponin Function at High Ca^2+^, but Does Not Reverse It, Provoking Strong Actin–Myosin Binding throughout the ATPase Cycle

It is known that omecamtiv mecarbil (OM) binds directly to cardiac myosin [49] and fastens the transient of actin–myosin from weak to strong-binding states [50]. In our model fibers which contain skeletal myosin heads, omecamtiv mecarbil also affects the conformational state of actin–myosin complex. Binding of 20 μM OM to myosin heads induces changes in the values of Φ_E_, ε, and N for S1-AEDANS both at high and low Ca^2+^ (Figure 6).

In the muscle fiber containing R168H-mutant tropomyosin in thin filaments at a high Ca^2+^, OM decreases the values of Φ_E_ and N, which indicates the rotation of the SH1 helix to the axis of the muscle fiber, a decrease in the mobility of probes, and hence the immobilization of myosin heads and the formation of a conformation of myosin heads characteristic for strong binding to actin. Since the mutation in tropomyosin inhibits the transition of myosin heads to a strong-binding form at high Ca^2+^ in the absence of nucleotide and in the presence of ADP (Figure 2 and Figure 3), the effect of OM can be considered restorative. However, when modeling the relaxation stage (in the presence of ATP), OM leads to a sharp decrease in the value of Φ_E_ to 43.8° at high Ca^2+^ and 42.5° at low Ca^2+^ and thus inhibits the conformational changes in the weak-to-strong transition of myosin heads (the value of Φ_E_ decreases to 0.8° versus 4.4° for the control in the absence of OM and 8.4° for the presence of wild-type tropomyosin, Figure 6a,d). Inhibition of the transition of myosin heads associated with OM to a state of weak binding to actin disrupts the correct cycle of the cross-bridge and its efficiency. In addition, OM significantly reduces the maximum ATPase activity of myosin (Figure 4), which correlates with a sharp decrease in the efficiency of myosin heads in the ATPase cycle.

At low Ca^2+^, OM significantly reduces the bending stiffness for muscle fibers containing R168Tpm as compared with muscle fibers containing WTTpm (Figure 6b,e). Such a decrease in the value of ε indicates a decrease in the persistence length of actin and an increase in this parameter for tropomyosin upon binding of myosin heads containing OM, which is typical for the activation of thin filaments. However, such changes become irrelevant in the presence of ATP and in the imitation of the relaxed state of the contractile system. As a result, a decrease in Ca^2+^ sensitivity at high Ca^2+^ and a decrease in the efficiency of cross-bridges in the ATPase cycle are observed (Figure 6d–f). Therefore, omecamtiv mecarbil is able to partially attenuate muscle tissue dysfunction caused by mutations in TPM3, but does not reverse it, provoking strong actin–myosin binding throughout the ATPase cycle.

### 2.6. The Troponin Inhibitor, EGCg, Restores Troponin Function at High Ca^2+^, the Ability of Muscle Tissue to Relaxation, and Weakly Restore the Ability of Troponin to Switch Actin Monomers off at Low Ca^2+^

Since the structural and functional state of proteins in the contractile system is interdependent, such reagents as specific modulators of Ca^2+^-sensitivity of thin filaments can at least partially restore the damage to the regulation mechanisms of Tpm and the effective cycling of myosin cross-bridges caused by the R168H mutation.

To desensitize myofilaments when Ca^2+^ levels are low, we tried to use the troponin inhibitor epigallocatechin-3-gallate (EGCg) [51]. Epigallocatechin-3-gallate (EGCg) is one of the catechins found in green tea (Camellia sinensis) in the highest quantity. Recent epidemiological evidence has shown that green tea consumption may be associated with a reduction in cardiovascular disease. It was found that EGCg binds to the C-domain of cardiac troponin C and has an inhibitory effect at concentrations of 25–100 μM: it reduces the Ca^2+^-sensitivity of cardiac myofilaments, the ATPase activity of actomyosin, and the maximum strength in skinned cardiac fibers. It is interesting to note that at a high concentration, EGCg potentiates contractile force and acts as a Ca^2+^-sensitizer pimobendan in guinea pig hearts [52], although it is considered a troponin inhibitor on its own.

A few facts are known about the hypotensive effect of EGCg and its positive effect on cardiac function, but there is no data on its effect on the contractile apparatus of the skeletal muscle. In the literature, there are reports that EGCg does not bind to skeletal muscle troponin C and has almost no effect on ATPase activity in myofibrils from skeletal muscle fibers [53]. At the same time, EGCg was found to significantly affect the Ca^2+^-dependent actin–S1–ATPase activity of reconstituted thin filaments with rabbit skeletal myosin and the potential to restore disorders associated with the presence of HCM-mutant troponin in thin filaments with this compound, provoking an increase in Ca^2+^ sensitivity [54].

In the system we assembled based on a ghost fiber, with thin filaments reconstructed from recombinant Tpm3.12 and skeletal muscle troponin, containing fluorescently labeled muscle skeletal S1, a strong effect of EGCg on the parameters of the polarized fluorescence of the probes was observed (Figure 7). No background signal was found that EGCg could create when adding any of the proteins—tropomyosin, troponin, or S1. The maximum fluorescence spectra of the probes associated with S1 in the absence and in the presence of EGCg did not differ significantly from each other. Therefore, EGCg is able to induce changes in the conformational rearrangements of the actin–Tpm–TN–S1 protein complex when modeling the functional states of thin filaments and stages of the ATPase cycle. The mechanism of action of EGCg in the skeletal muscle contractile apparatus remains to be discovered. In the present work, only the possibility of changing the conformational rearrangement of myosin heads disturbed by the presence of the R168H-mutant tropomyosin was studied.

Changes induced by EGCg are shown for the complex of proteins at low and high calcium ion concentrations. In the absence of nucleotide, the R168H mutant increases the Φ_E_ angle at high Ca^2+^ concentrations (Figure 2a), whereas 30 μM EGCg causes a decrease in this parameter (Φ_E_ value decreases by 0.4°, Figure 7a). Thus, if R168HTpm reduces the number of those myosin heads that are in a conformation of strong binding to actin, then EGCg, to some extent, cancels the effect of the mutation. In the presence of ATP, EGCg increases the values of Φ_E_ and N, i.e., the number of heads capable of relaxation increases. Therefore, during the weak-to-strong transition, the amplitude of changes in Φ_E_ restores from 1.2° (in the presence of the R168H mutant) to 2.4° (in the presence of the R168H mutant and EGCg). However, EGCg was not found to significantly increase the value of Φ_E_ for the state in the absence of nucleotide at a low level of Ca^2+^, which is always observed by us in the presence of control tropomyosin (Figure 2a). That is, it does not cause pronounced Ca^2+^-sensitive changes in the pool of strongly and weakly bound myosin heads, although a trend towards this is observed (the values of Φ_E_ are 45.0° at high Ca^2+^ and 45.3° at low Ca^2+^ in the presence of EGCg, versus 43.6° and 45.3°, respectively, in the system containing WTTpm), and this pattern intensifies in the presence of MgADP.

However, EGCg continues to increase the number of strongly bound myosin heads at low Ca^2+^ (Φ_E_ values are equal for ±EGCg) and reduces the work of cross-bridges (Figure 8). The amplitude of changes in the values of Φ_E_ at the transition from the state in the presence of MgATP to the state in the absence of nucleotides for R168HTpm–EGCg was less by 7.3° than for WTTpm and by 4.5° than for R168HTpm–Tir (see below). This means that EGCg weakly rehabilitates the efficiency of the cross-bridges work and is thus able to partially correct the disturbance induced by the R168H mutation. Therefore, epigallocatechin-3-gallate can probably be used to alleviate muscle dysfunction.

### 2.7. The Troponin Ca^2+^-Sensitive Activator, Tirasemtiv, Can Correct the Dysregulation Induced by the R168H Mutation in Tpm

It is known that tirasemtiv (Tir) binds specifically with high affinity to TN but does not interact with actin, myosin, or Tpm [55], so it can be used as a specific activator of calcium activation in skinned skeletal muscle fibers [56,57,58]. In addition, it was previously shown that the troponin activator CK-2066260, a structural analog of tirasemtiv, can correct the low calcium sensitivity of sarcomeres induced by the point mutation [59]. Here, we tried to use tirasemtiv to reduce the disruption of the actin–myosin interaction caused by the R168H mutation in Tpm3.12 during the ATPase cycle in ghost muscle fibers.

Binding of 5 μM Tir to the F-actin–R168HTpm–TN–S1-AEDANS complex induces a change in the values of Φ_E_, ε, and N both at high and low Ca^2+^ (Figure 9).

In the absence of a nucleotide at high Ca^2+^, the addition of 5 μM tirasemtiv increased the amount of myosin heads strongly bound to F-actin (Φ_E_ values decreased by 0.9°), while bending stiffness (ε) remained practically unchanged, and flexibility of the myosin head attachment to F-actin (N) decreased by 0.052 rel. units. A similar effect was also observed in the presence of MgADP (Figure 9). This means that Tir restores the ability of myosin heads to bind strongly to F-actin during the ATPase cycle at high Ca^2+^. In addition, Tir, in the presence of MgATP, increased the values of Φ_E_ and ε, showing that the mutation can increase the efficiency of the cross-bridge work. Indeed, the amplitude of change in the value of Φ_E_ (at the transition from MgATP to no nucleotides) for R168HTpm–Tir was 6.9°, which is higher than the amplitude observed for R168HTpm (5.8°) (Figure 2a and Figure 9a).

At low Ca^2+^, tirasemtiv slowly increased the number of myosin heads strongly bound to F-actin (the values of Φ_E_ increased in the absence of nucleotides and in the presence of MgADP by 0.4° and 0.6°, respectively) (Figure 9a). In the absence of nucleotides, ε, practically nothing changed. In the presence of tirasemtiv, the flexibility of myosin head attachment to F-actin (N) decreased by 0.055 rel. units when mimicking the AM state and did not change in the presence of MgADP. In the presence of MgATP, Tir increased the values of Φ_E_ (at low Ca^2+^) and ε (at high Ca^2+^) showing a decrease in the amount of switched-on actin monomers and strongly bound myosin heads (Figure 9a,b).

Thus, tirasemtiv partially restores the ability of troponin to increase the relative number of myosin heads strongly associated with actin at high Ca^2+^ (Figure 9, Figure 10 and Figure 11), which allows us to hopefully reduce the effect of the mutation on the ability of the contractile system to generate tension. In the presence of Tir, the effect of the mutation on the functioning of tropomyosin slightly changes at low Ca^2+^ (in the presence of MgADP). Troponin, instead of switching actin monomers off and reducing the formation of a strong form of actin–myosin interaction, continues to increase the Ca^2+^ sensitivity of thin filaments (Figure 2, Figure 9 and Figure 11). This is consistent with the data obtained in the protein solution showing that tirasemtiv additionally increases the Ca^2+^-sensitivity of thin filaments containing the mutant Tpm (Figure 4) and, therefore, can significantly stimulate the actin-activated ATPase activity of myosin in the highly ordered contractile system within the muscle fiber. Therefore, tirasemtiv is likely to be used to alleviate muscle tissue dysfunction.

## 3. Materials and Methods

### 3.1. Using Experimental Animals

Muscle fibers and proteins were extracted from the skeletal muscles of rabbits (*Oryctolagus cuniculus*). The animals were killed in accordance with the official regulations of the community council on the use of laboratory animals [8,9,10,11]. The study was approved by the Animal Ethics Committee of the Institute of Cytology of the Russian Academy of Science (Assurance Identification Number F18-00380, valid until 31 October 2022).

### 3.2. Preparation of Proteins and Their Labeling by Fluorescent Probes

Skeletal muscle myosin and troponin were isolated and purified using standard protocols [60,61]. Treatment of myosin with α-chymotrypsin (Sigma–Aldrich, St. Louis, Missouri, USA) for 20 min at 25 °C yielded myosin subfragment-1 (S1) free of the regulatory light chains [62]. S1 was modified at Cys707 with 1,5-IAEDANS (Molecular Probes, Eugene, Oregon, USA), as described earlier [63]. Recombinant wild-type γγ-Tpm (a control protein without mutations) and the R168H mutant were obtained using molecular genetics methods, as described previously [39,64]. Briefly, human Tpm3.12 cDNA (Oxford University, UK) was amplified by polymerase chain reaction (PCR) to introduce the mutation. The product was cloned into the bacterial expression vector pMW172 using NdeI and HindIII. Nine base pairs encoding Met–Ala–Ser at the N-terminus of Tpm were added to the construct to compensate for the reduced affinity of recombinant non-acetylated skeletal Tpm for F-actin. The Tpm sequence was altered using a two-step polymerase chain reaction-based oligonucleotide-directed mutagenesis protocol to produce the R168H substitution. The PCR products were cloned and sequenced to verify the substitution. Then, the constructs were overexpressed in *Escherichia coli* strain BL21(DE3)pLysS according to standard methods. Bacterial cell lysates containing Tpms were heated to 90 °C before clarification by centrifugation at 33,000× *g* for 20 min. The resulting supernatant was fractionated by reducing the pH to 4.8. Both wild-type and mutant Tpms were purified from the extract using anion exchange chromatography. The prepared Tpms were stored at −45 °C for several months. The purity of the proteins was examined by SDS–PAGE (Figure 1).

### 3.3. Determination of Actin-Activated ATPase of Subfragment-1

The rate of the ATPase reaction was determined for fully regulated reconstituted thin filaments in a solution containing 1 μM S1, 7 μM F-actin, 3 μM troponin, 3 μM WTTpm or R168HTpm in the following buffer: 12 mM Tris-HCl (pH 7.9), 2.5 mM MgCl_2_, 15 mM KCl, 20 mM NaCl, 0.2 mM dithiothreitol, and 2 mM ATP at 25 °C (Sigma–Aldrich, St. Louis, Missouri, USA). The reaction was carried out by increasing Ca^2+^ concentrations from 1 × 10^−9^ M to 1 × 10^−4^ M. The concentration of free Ca^2+^ in the presence of 2 mM EGTA (Sigma–Aldrich, St. Louis, Missouri, USA) was calculated using the Maxchelator program (http://maxchelator.stanford.edu/CaEGTA-TS.htm, accessed on 1 November 2019). The reaction was stopped after 10 min by adding trichloroacetic acid (LenReactiv, Saint Petersburg, Russia) to a final concentration of 5%. The amount of inorganic phosphate formed was determined by the method of Fiske and Subbarow [65]. Three experiments were conducted for each experimental condition. Statistical processing of data, calculation of the pCa_50_ value, and plotting were carried out using the GraphPad Prism 5.0 software.

### 3.4. Preparation and Labeling of Ghost Fibers

Models of striated muscle fibers, in which, due to the extraction of myosin and the regulatory proteins, actin comprises up to 90% of the total muscle protein, were used in this work. These models (so-called ghost fibers) were obtained from *M. psoas* of rabbits. Bundles of about 100 fibers were placed into a cooled solution containing 100 mM KCl, 1 mM MgCl_2_, 67 mM K, Na phosphate buffer, pH 7.0 (Sigma–Aldrich, St. Louis, Missouri, USA), and 50% glycerol (PanReac Applichem, Darmstadt, Germany). Single fibers were gently isolated from the glycerinated muscle bundle and incubated for 70–80 min in a solution containing 800 mM KCl, 1 mM MgCl_2_, 10 mM ATP, 6.7 mM K, Na phosphate buffer, pH 7.0 [8,10]. Thin filaments were reconstructed with Tpm (WTTpm or R168HTpm) and troponin and decorated with S1 by incubating the fiber in a solution containing 50 mM KCl, 3 mM MgCl_2_, 1 mM dithiothreitol, 6.7 mM K, Na phosphate buffer, pH 7.0, and the corresponding proteins. Proteins that did not bind to F-actin were removed by washing the fiber in the same protein-free solution.

The final composition of the fibers was examined using 12% SDS–PAGE gels, stained with Coomassie brilliant blue R (Sigma–Aldrich, St. Louis, Missouri, USA), and scanned using Bio–Rad ChemiDocTM MP Imaging system (Hercules, CA, USA) (Figure 1). Then, 8–10 fibers were applied to each lane. Excess of the proteins was removed by 60 min flushing of the fibers in the washing solution, which contained 100 mM KCl, and 1 mM MgCl_2_, 67 mM K, Na phosphate buffer, pH 7.0. The ratio of WTTpm to the mutant Tpm that bound to actin was determined by Image Lab software, version 6.0.0 (Bio–Rad, Hercules, California, USA).

### 3.5. Polarized Fluorescence Measurements

Steady-state polarized fluorescence was measured in ghost fibers using a flow-through chamber and a polarized fluorimeter, as described before [32]. The fluorescence from the 1,5-IAEDANS-labeled S1 was excited at 407 ± 5 nm. Fluorescence intensity (I) was recorded in the range of 500–600 nm. Probes in ghost fibers were excited by a 250 W mercury lamp DRSH–250 [10]. Exciting light passed through a quartz lens and a double monochromator and split into two polarized beams by a polarizing prism. The ordinary polarized beam was reflected on a dichroic mirror and condensed with a quartz objective (UV 58/0.80) on a fiber in a cell on the microscope stage. The light emitted by the fiber was collected by the objective and led to a concave mirror with a small hole. After passing through the lens and the barrier filter, the beam was separated by a Wollaston prism into polarized beams perpendicular and parallel to the fiber axis. The intensities of the four polarized fluorescence components _‖_I_‖_, _‖_I_⊥_, _⊥_I_⊥_ and _⊥_I_‖_ were detected by two photomultiplier tubes [10,32]. Fluorescence polarization ratios were defined as: P_‖_ = (_‖_I_‖_ − _‖_I_⊥_)/(_‖_I_‖_ + _‖_I_⊥_) and P_⊥_ = (_⊥_I_⊥_ − _⊥_I_‖_)/(_⊥_I_⊥_ + _⊥_I_‖_). The subscripts ‖ and ⊥ designate the direction of polarization, parallel and perpendicular to the fiber axis; the former denotes the direction of polarization of the incident light, the latter that of the emitted light. In all experiments, the background fluorescence intensity of the ghost fiber was 2–3 orders of magnitude lower than the fluorescence intensity of the probe specifically associated with the protein and was taken into account in data processing.

The experimental data were assessed using the helix-plus isotropic model [32,66,67]. The model is based on the assumption that there are two populations of fluorophores in a muscle fiber: ordered fluorophores in the amount (1–N), whose absorption and emission oscillators are oriented at the angles Φ_A_ and Φ_E_, respectively, relative to the thin filament axis, and disordered fluorophores in the amount N (oriented under the magic angle 54.7°). The number of disordered probes (N) is related to the mobility of the labeled protein. Motions of the probes relative to the protein are included in the model as the angle γ (the angle between the absorption and emission dipoles). The value of γ is constant for 1,5-IAEDANS bound to S1 [10,32]. In this model, the thin filament is assumed to be flexible, i.e., the angle θ between the fiber axis and the thin filament is not zero. According to the theory of a semiflexible filament, for a filament length L with one end fixed and the other end free, sin^2^θ = 0.87(kT/ε)L. Thus, the bending stiffness (ε) of actin filaments can be estimated from sin^2^θ [66].

The measurements were carried out in washing solution in the absence of nucleotides (simulating the AM state of the actomyosin complex) or in the presence of 3 mM ADP or 5 mM ATP, mimicking, respectively, the AM^•ADP and AM*•ATP states of actomyosin in the ATPase cycle [8,68]. In experiments with troponin, the solutions additionally contained either 0.1 mM CaCl_2_ or 4 mM EGTA. The effect of the troponin and myosin modulators was examined in addition to the fibers: 5 μM tirasemtiv (Calbiochem, Merck KGaA, Darmstadt, Germany), 20 mM 2,3-butanedione monoxime (Sigma–Aldrich, Merck KGaA, Darmstadt, Germany), 20 μM omecamtiv mecarbil (Axon Medchem, Groningen, Netherlands), or 30 μM epigallocatechin-3-gallate (Sigma–Aldrich, Merck KGaA, Darmstadt, Germany).

Changes in polarized fluorescence parameters (Φ_E_, ε, and N) were considered to report on conformational changes in the protein modified with the probe [8,9,10,11]. The data were obtained from 4−6 fibers (20–30 measurements) for each experimental condition. The statistical significance of changes in two samples for each experiment (mutant and wild-type tropomyosin, or in the absence and in the presence of tirasemtiv) was evaluated using Student’s *t*-test, *p* < 0.05.

## 4. Conclusions

Summing up, the major advantage of our in situ structural approach compared to previous studies of the regulation of the actin–myosin interaction in protein solution using isolated filaments is that tropomyosin orientation was determined under physiological conditions and in an intact muscle sarcomere, preserving the native relationship between myosin and actin filaments. The application of reconstituted muscle fibers enabled us to reveal unknown details of the regulation of the actin–myosin interaction by the tropomyosin–troponin complex during the ATPase cycle in muscle fibers containing wild-type and mutant R168H Tpms. Our data showed that Ca^2+^ regulation of the actin–myosin interaction is mediated by conformational changes in the tropomyosin–troponin complex, actin, and myosin heads. This leads to spatial rearrangements and alterations in the persistence length of Tpm and F-actin, which presumably cause an azimuthal shifting of the tropomyosin strands. Conformational changes in the tropomyosin–troponin complex, F-actin, and myosin heads initiated by Ca^2+^ and nucleotides are interdependent [8,9,10,11], so a point mutation in any of these proteins should disrupt this interdependence and may induce deregulation of the actin–myosin interaction. Our work demonstrates that the R168H substation induces such uncoupling. Indeed, troponin loses its ability to move the Tpm strands to the outer domain of actin and switch actin monomers off at low Ca^2+^ (Figure 2 and Figure 3). This may contribute to the high Ca^2+^ sensitivity that we observed in the protein solution (Figure 4) and was found previously [33]. In addition, the R168H mutation can also alter the ability of Tpm to control the formation of strong binding of myosin heads to F-actin throughout the ATPase cycle; the amount of the myosin heads strongly bound to F-actin decreases at mimicking the AM and AM^•ADP stages (Figure 2 and Figure 3); therefore, the actin-activated ATPase activity of myosin in skeletal muscle tissue can decrease, and muscle weakness is observed [8,10,11,39,40,41,42].

We suggested that the replacement of positively charged arginine 168 with positively charged histidine changes the tropomyosin–troponin interaction and, as a result, partially destabilizes the tropomyosin molecule in the region of the site for Tpm binding to TN. The alteration between Tpm and TN can disrupt troponin function by interfering with its ability to switch the thin filaments on and off. This can lead to inhibition of ATPase activity at high Ca^2+^ (a decrease in force production) and an increase in Ca^2+^ sensitivity, as well as the appearance of the rigor-like myosin heads that are strongly bound to F-actin during relaxation (Figure 2, Figure 3 and Figure 4). Rigor-like cross-bridges can induce contracture and disorganization of muscle fiber [26]. Therefore, it seems important to reduce the effect of the R168H substitution, for which we used several modulators of troponin and myosin functions (Figure 5, Figure 6, Figure 7, Figure 8, Figure 9, Figure 10, Figure 11 and Figure 12). Only two of them—the fast skeletal troponin activator tirasemtiv and the troponin inhibitor epigallocatechin-3-gallate—have been shown to restore the ability of troponin to activate the strong binding of the myosin heads to F-actin at high Ca^2+^ and reduce the amount of rigor-like myosin heads during relaxation. Therefore, tirasemtiv and epigallocatechin-3-gallate may be more likely to be used to relieve muscle tissue dysfunction and reduce muscle weakness caused by the R168H mutation.

## Figures and Tables

**Figure 1 ijms-24-05829-f001:**
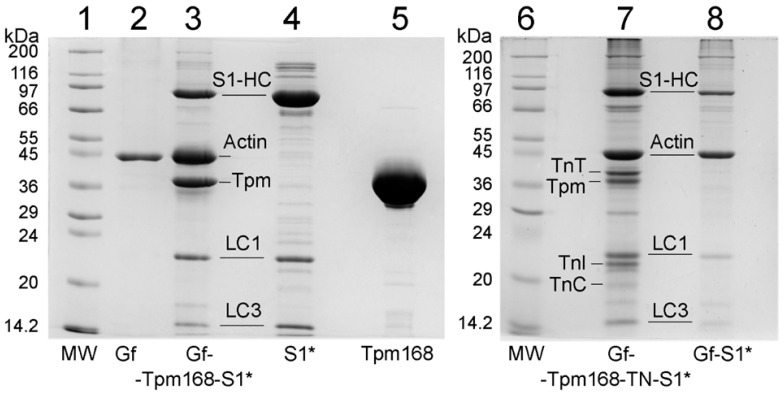
Control of protein purity and muscle fiber composition. Proteins: S1-AEDANS, lane 4; R168HTpm (Tpm168), lane 5. Muscle fibers: ghost fibers, lane 2; ghost fibers containing S1-AEDANS, lane 8; ghost fibers with thin filaments reconstructed by the R168H-mutant tropomyosin and containing S1-AEDANS in the absence of troponin, lane 3, and in the presence of troponin (TN), lane 7. Unbound proteins were removed by exposing the fibers to the washing solution for 15 min. Here, 6–8 fibers were used to prepare a probe loaded into a gel. As calculated using ImageLab 6.0, ghost fibers (lane 2) were composed of F-actin by 93.6%. The ratio of the R168H-mutant Tpm bound to actin was as high as for the Tpm control [10].

**Figure 2 ijms-24-05829-f002:**
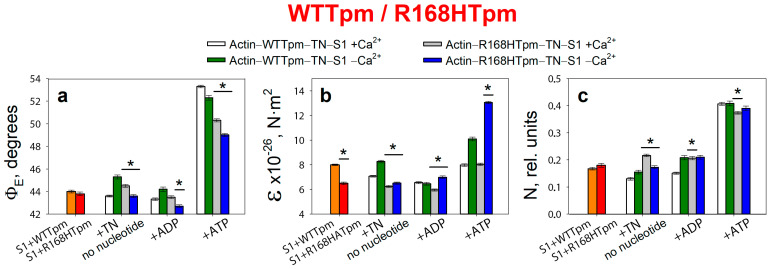
The effect of troponin (TN) and Ca^2+^ on the values of Φ_E_ (**a**), ɛ (**b**), and N (**c**) of polarized fluorescence of S1-AEDANS in the presence of WTTpm or R168HTpm was revealed in ghost fibers under conditions of simulating the sequential steps of the actomyosin ATPase cycle. The data are averages for 4–6 fibers for each experimental condition. The data are presented as mean ± SEM. * *p* < 0.05 = difference between WTTpm and R168HTpm.

**Figure 3 ijms-24-05829-f003:**
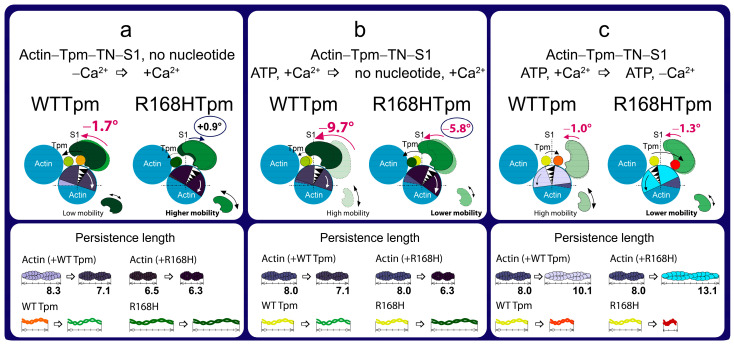
The scheme explains the effect of the R168H mutation on the spatial rearrangement of the myosin head (S1), the position of tropomyosin (Tpm), the angular orientation of actin monomer, and the persistence lengths of the Tpm strands and actin filaments. Three different transitions are shown, influenced by (**a**) an increase in Ca^2+^ concentration in the absence of nucleotides, (**b**) a weak-to-strong transition, and (**c**) a decrease in Ca^2+^ concentration in the presence of ATP. Information was obtained from the calculation of the value of Φ_E_—the orientation angle of the emission dipoles of the fluorescent probes bound to S1, and the value of ε—the bending stiffness (shown here as the alterations in the persistence lengths of the Tpm strands and actin filaments). The N parameter was used to judge the increase or decrease in the mobility of myosin heads (shown by big and small double-edged arrows). In all states, except for the high Ca^2+^ in the absence of the nucleotide, the conformation of S1 in the presence of R168HTpm is characteristic of the formation of strong binding of myosin to actin. The mutation reduces the persistence length of actin in the absence of the nucleotide and significantly increases it in the presence of ATP at low Ca^2+^. The persistence length of Tpm changes conversely. The R168H substitution prevents the activation of the protein complex in response to an increase in the Ca^2+^ concentration. Designations: numbers show changes in the value of Φ_E_ for S1-AEDANS between states (red numbers are characteristic of an increase in the population of strong-binding myosin heads). Arrows show directions of tilt of the myosin head and rotation of actin monomer relative to the fiber axis, as well as displacement of tropomyosin over actin. Different localizations of Tpm and conformational states of actin and their corresponding persistence lengths are depicted by different colors.

**Figure 4 ijms-24-05829-f004:**
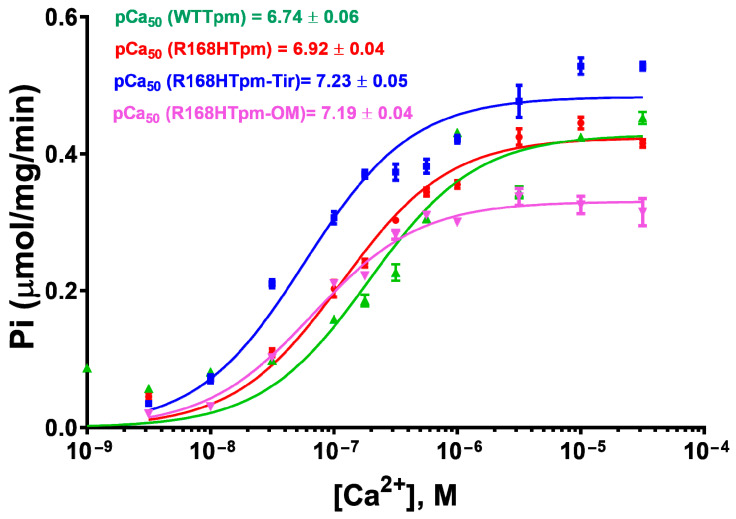
Effect of the R168H mutation in γ-tropomyosin (Tpm) on the sensitivity of thin filaments to Ca^2+^ activation. Ca^2+^ dependence was determined for fully regulated reconstituted thin filaments. Actin–S1 ATPase was measured in the presence of wild-type (WT) Tpm (denoted by triangular symbols) and R168H-mutant Tpm (round symbols) at 25 °C. The effects of two chemical compounds were shown—tirasemtiv (Tir, squire symbols) and omecamtiv mecarbil (OM, inverted triangles). Error bars indicate ± SEM. The pCa values were calculated from the data averaged from three experiments. Conditions are given in Materials and Methods. ATPase assay revealed that the mutation increases the Ca^2+^ sensitivity of thin filaments. The midpoints of the curves (pCa_50_) are 6.74 ± 0.06 for filaments containing WTTpm and 6.92 ± 0.04 for those reconstituted with R168HTpm. Tir (5 μM) and OM (20 μM) increased the value of pCa_50_ to 7.23 ± 0.05 and 7.19 ± 0.04, respectively (*p* < 0.01).

**Figure 5 ijms-24-05829-f005:**
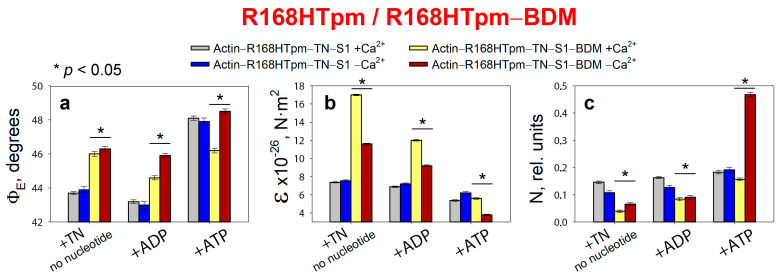
Effect of 20 mM 2,3-butanedione monoxime (BDM) on the values of Φ_E_ (**a**), ɛ (**b**), and N (**c**) of polarized fluorescence of S1-AEDANS in the presence of R168HTpm in ghost muscle fibers under conditions simulating the sequential steps of the actomyosin ATPase cycle. The results were obtained in the same experiment when the muscle fibers containing the R168HTpm without BDM were used as a control. The data are presented as mean ± SEM. * *p* < 0.05 = difference between R168HTpm and R168HTpm–BDM.

**Figure 6 ijms-24-05829-f006:**
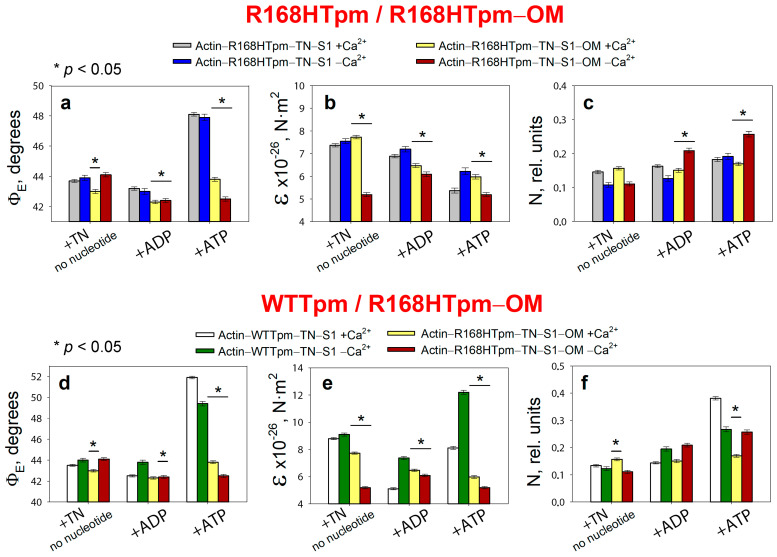
Effect of 20 μM omecamtiv mecarbil (OM) on the values of Φ_E_ (**a**), ɛ (**b**), and N (**c**) of polarized fluorescence of S1-AEDANS in the presence of WTTpm or R168HTpm revealed in ghost muscle fibers under conditions simulating the sequential steps of the actomyosin ATPase cycle. The comparison of the effect of OM is given for two series of experiments, when the muscle fibers containing the R168HTpm (**a**–**c**) or WTTpm (**d**–**f**) without OM were used as a control. The data are presented as mean ± SEM. * *p* < 0.05 = difference between R168HTpm and R168HTpm–OM (**a**–**c**), and WTTpm and R168HTpm–OM (**d**–**f**).

**Figure 7 ijms-24-05829-f007:**
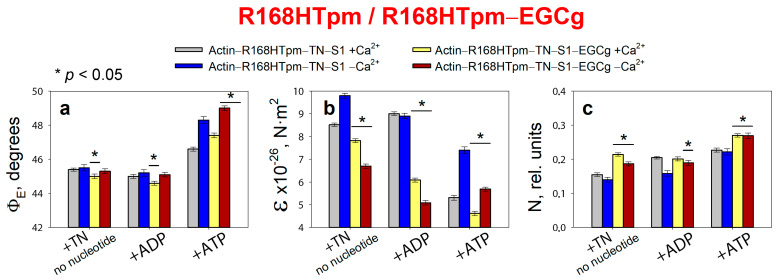
Effect of 30 μM epigallocatechin-3-gallate (EGCg) on the values of Φ_E_ (**a**), ɛ (**b**), and N (**c**) of polarized fluorescence of S1-AEDANS in the presence of R168HTpm. The changes are revealed in ghost muscle fibers under conditions simulating the sequential steps of the actomyosin ATPase cycle. The fluorescence parameters presented here are compared with the data obtained in the same experiment when the muscle fibers containing the R168HTpm without EGCg were used as a control. The data are presented as mean ± SEM. * *p* < 0.05 = difference between R168HTpm and R168HTpm–EGCg.

**Figure 8 ijms-24-05829-f008:**
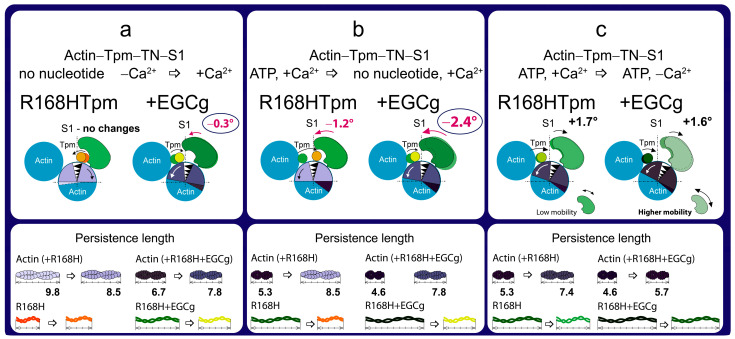
The scheme explains the effect of the troponin inhibitor EGCg on the spatial rearrangement of myosin head, the position of the R168H-mutant tropomyosin (Tpm), the angular orientation of actin monomer, and the persistence lengths of the Tpm strands and actin filaments compared to R168HTpm in the absence of EGCg. Three different transitions are shown, influenced by (**a**) an increase in Ca^2+^ concentration in the absence of nucleotides, (**b**) a weak-to-strong transition, and (**c**) a decrease in Ca^2+^ concentration in the presence of ATP. The designations are as in Figure 3.

**Figure 9 ijms-24-05829-f009:**
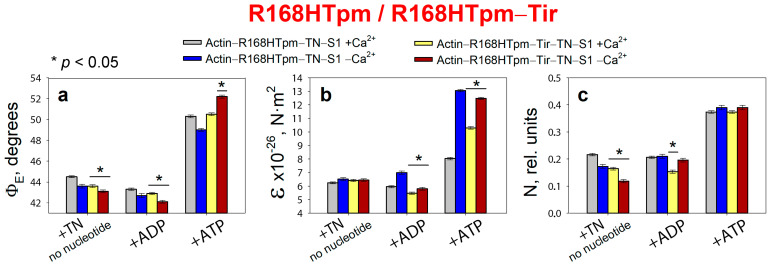
Influence of 5 μM tirasemtiv (Tir) on the values of Φ_E_ (**a**), ɛ (**b**), and N (**c**) of the polarized fluorescence of S1-AEDANS, in the presence of R168HTpm, revealed in ghost muscle fibers under conditions simulating the sequential steps of the actomyosin ATPase cycle. The experimental conditions and designations are described in Materials and Methods. The fluorescence parameters presented here are compared with the data obtained in the same experiment when the muscle fibers containing the R168HTpm without Tir were used as a control. The data are averages for 4–6 fibers for each experimental condition. The data are presented as mean ± SEM. * *p* < 0.05 = difference between R168HTpm and R168HTpm–Tir.

**Figure 10 ijms-24-05829-f010:**
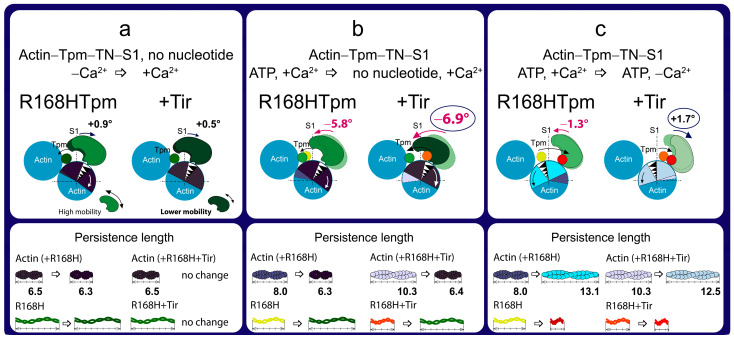
The scheme explains the effect of the fast skeletal troponin activator tirasemtiv (Tir) on the spatial rearrangement of myosin head, the position of the R168H-mutant tropomyosin (Tpm), the angular orientation of actin monomer, and the persistence lengths of the Tpm strands and actin filaments. Three different transitions are shown, influenced by (**a**) an increase in Ca^2+^ concentration in the absence of nucleotides, (**b**) a weak-to-strong transition, and (**c**) a decrease in Ca^2+^ concentration in the presence of ATP. The R168H substitution prevents the activation of the protein complex in response to an increase in Ca^2+^ concentration and inhibits relaxation; tirasemtiv allows activation as well as relaxation. The amplitude of the change in the value of Φ_E_ at the transition between states in the presence of MgATP and in the absence of a nucleotide is increased by Tir. The designations are as in Figure 3.

**Figure 11 ijms-24-05829-f011:**
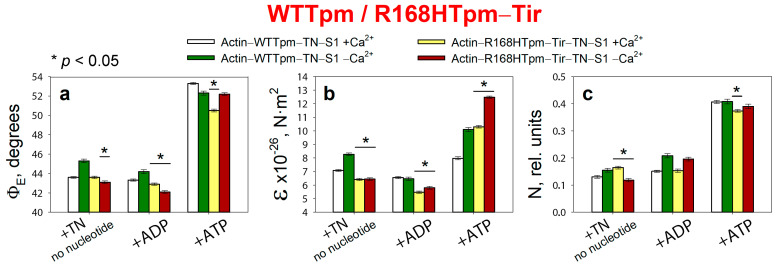
Effect of 5 μM tirasemtiv (Tir) on the values of Φ_E_ (**a**), ɛ (**b**), and N (**c**) of polarized fluorescence of S1-AEDANS in the presence of WTTpm or R168HTpm revealed in ghost muscle fibers under conditions simulating the sequential steps of the actomyosin ATPase cycle. The fluorescence parameters presented here are compared with the data obtained in the same experiment when the muscle fibers containing the WTTpm were used as a control. The data are presented as mean ± SEM. * *p* < 0.05 = difference between WTTpm and R168HTpm–Tir.

**Figure 12 ijms-24-05829-f012:**
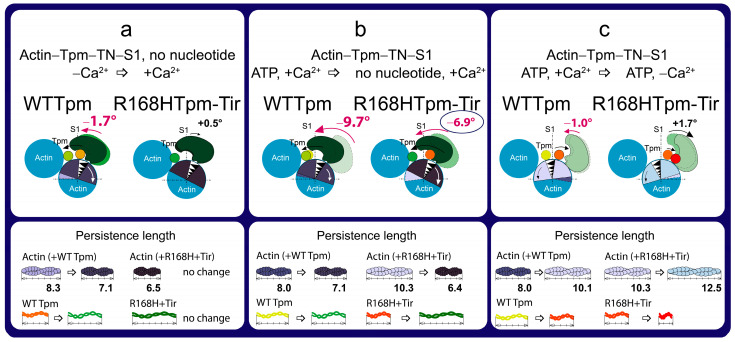
The scheme explains the effect of the fast skeletal troponin activator tirasemtiv (Tir) on the spatial rearrangements of myosin head, the position of the R168H-mutant tropomyosin (Tpm), the angular orientation of actin monomer, and the persistence lengths of the Tpm strands and actin filaments compared to WTTpm. Three different transitions are shown, influenced by (**a**) an increase in Ca^2+^ concentration in the absence of nucleotides, (**b**) a weak-to-strong transition, and (**c**) a decrease in Ca^2+^ concentration in the presence of ATP. Tirasemtiv normalizes the state at high Ca^2+^ in the absence of a nucleotide and at low Ca^2+^ in the presence of MgATP. Tirasemtiv does not normalize the state at low Ca^2+^ in the absence of a nucleotide and at high Ca^2+^ in the presence of MgATP. The designations are as in Figure 3.

## Data Availability

Not applicable.

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
