# Peer review of "Molecular Mechanisms of Deregulation of Muscle Contractility Caused by the R168H Mutation in TPM3 and Its Attenuation by Therapeutic Agents"

_ijms, 2023, doi:10.3390/ijms24065829_

Round 1
Reviewer 1 Report
In this paper the authors aimed to characterize the pathogenic R168H mutation of TPM3, a muscle specific tropomyosin isoform, linked to congenital muscle fiber type disproportion (CFTD) and muscle weakness. In particular, they focused on understanding the molecular mechanisms leading to muscle dysfunction. As a main model system they used a reconstituted muscle fiber system to study the effect of the R168H mutation on the actin-myosin interaction in different conditions (e.g. in the presence of low or high Ca2+ ion concentration). They show that the R168H mutation of TPM3 allows a stronger binding of the myosin heads to F-actin at low Ca2+ as compared to wild type, whereas at high Ca2+ level it reduces the binding strength between myosin and actin. In addition, they also show that the effect of the mutant TPM3 is mediated by troponin, because in the presence of the R168H mutant form at low Ca2+, troponin switches the actin monomers on and erroneously activates strong actin-myosin binding. Conversely, at high Ca2+, troponin instead of activating the formation of strongly bound myosin heads, reduces the amount of switched-on actin monomers and decreases the relative number of myosin heads strongly bound to actin. Thus the R168H mutation causes an abnormally high Ca2+-sensitivity on the one hand, and parallel to that, it reduces the efficiency of the cross-bridge work. The higher Ca2+ sensitivity is likely to result in partial inhibition of muscle fiber relaxation, which together with the decrease in work-production, will lead to muscle weakness. Grounded on these findings, the authors tested four commonly used drugs whether they can restore the contractility of the R168H mutant muscle fibers. These studies led to the conclusion that the fast skeletal troponin activator tirasemtiv, and to a lesser extent, the troponin inhibitor epigallo-catechin-3-gallate can restore the ability of troponin to activate the strong binding of the myosin heads to F-actin at high Ca2+, and reduce the amount of rigor- like myosin heads during relaxation.
Overall, I find this work carefully done and potentially suitable for publication. I have one major concern though, notably, that the ghost muscle fiber system is rarely used in other studies, therefore it would be more convincing if at least some of the most crucial findings are backed up by data generated in other model systems suitable to measure the parameters of the cross-bridge interaction. Beyond this, I have a number of remarks regarding formal issues:
lines 1-5: The title is too complicated, it should be shortened. Also, it contains a mistake as “Potential therapeutic agents” was left in their by mistaken (at least, in my understanding the title should be a single sentence)
lines 13-49: The Abstract is too long, it should be far more concise.
lines 210-211: This title does not make sense, it should be modified for a grammatically correct sentence.
lines 426-433: Instead of being here, these sentences should go into the introductory part of the next paragraph where they they really belong to.
lines 521-526: The same issue as above, these considerations would be much better placed at the beginning of the next session.
line 540 and 544: “Something is known” and “we also came across a paper” are phrases that read quite odd in a scientific paper. These should be avoided.
In addition, I noted a good number of other grammatical errors, and therefore correction of English usage is highly recommended. Moreover, some sentences are overly complicated, these should be rephrased.
Author Response
We are very grateful to the Reviewer for the detailed consideration of our work and find the comments fair and constructive. We have addressed each specific point raised by the Reviewer by making additions and clarifications to the text. Please see the attachment.
Prof. Yurii Borovikov, Dr. Olga Karpicheva

Reviewer 2 Report
The authors' present article investigated the mechanism by which Arg168 mutation in tropomyosin causes muscle dysfunction. As a related mechanism, the mutation was suggested to cause an abnormality in calcium-induced muscle contraction/relaxation control. This study provides new information on the functional regulation of the R168H mutation. However, it does suggest a few things to correct.
1. Since the title and abstract of the study are too complex and long, the reviewer requests that they be briefly rewritten based on key information.
2. In the comparison between groups in Figure 2, the insignificant is marked only with 'NS'. Indicate the p-value or symbol for significant differences between groups.
3. In all Figures, the significant differences between groups are indicated in text only in the figure legend. Please indicate the difference between groups as a p value or a symbol on the graph.
4. Describe in more detail the production of R168HTpm mutant ghost fiber muscle in the experimental methods and materials section.
Author Response

(The authors gave the same response as above.)
